# The Influence of Solution pH on the Kinetics of Resorcinol Electrooxidation (Degradation) on Polycrystalline Platinum

**DOI:** 10.3390/molecules24122309

**Published:** 2019-06-21

**Authors:** Tomasz Mikolajczyk, Mateusz Luba, Boguslaw Pierozynski, Ireneusz M. Kowalski, Wieslaw Wiczkowski

**Affiliations:** 1Department of Chemistry, Faculty of Environmental Management and Agriculture, University of Warmia and Mazury in Olsztyn, Plac Lodzki 4, 10-727 Olsztyn, Poland; mateusz.luba@uwm.edu.pl; 2Department of Rehabilitation, Faculty of Medical Sciences, University of Warmia and Mazury in Olsztyn, Zolnierska 14C Street, 10-561 Olsztyn, Poland; rehab@uwm.edu.pl; 3Institute of Animal Reproduction and Food Research, Polish Academy of Sciences in Olsztyn, Tuwima 10 Street, 10-748 Olsztyn, Poland; w.wiczkowski@pan.olsztyn.pl

**Keywords:** oxidation of resorcinol, a.c. impedance spectroscopy, energy dispersive X-ray spectroscopy, 0.5 M H_2_SO_4_, 0.5 M Na_2_SO_4_, 0.1 M NaOH, 0.1 M H_2_SO_4_, 0.5 M NaOH

## Abstract

Electrochemical oxidation of resorcinol on a polycrystalline platinum electrode was investigated in five different solutions, namely 0.5 and 0.1 M H_2_SO_4_, 0.5 M Na_2_SO_4,_ 0.5 and 0.1 M NaOH. The rates of electrochemical degradation of resorcinol were determined based on the obtained reaction parameters, such as resistance, capacitance and current-density. The electrochemical analyses (cyclic voltammetry and a.c. impedance spectroscopy) were carried-out by means of a three-compartment, Pyrex glass cell. These results showed that the electrochemical oxidation of resorcinol is strongly pH-dependent. In addition, the energy dispersive X-ray (EDX) spectroscopy technique was employed for Pt electrode surface characterization. Additionally, the quantitative determination of resorcinol removal was performed by means of instrumental high-performance liquid chromatography/mass spectrometry (HPLC/MS) methodology.

## 1. Introduction

Pollution of the natural environment has become one of the main topics of sustainable development. In fact, organic pollutants are associated with some of the most dangerous chemicals. A very good example of such a harmful substance is resorcinol, due to its severe toxicity, low biodegradability, as well as its widespread use in many technological processes. Resorcinol belongs to a group of phenolic compounds, which are classified as a toxic industrial pollutants with severe health consequences. Even at low concentrations, phenols can lead to thyroid dysfunction, causing damage to red blood cells and the liver. Moreover, this group of pollutants possesses carcinogenic potential and high reactivity. Thus, in the presence of microorganisms, both inorganic and organic compounds dissolved in water could produce other harmful substances [1,2,3,4]. The main source of resorcinol is industrial wastewaters, posing problems for such important industry sectors such as petrochemicals and drug manufacturing [5,6,7,8,9,10].

Nowadays, multiple techniques are used for removing pollutants from water and environment. Ozonation is a chemical water treatment technique, which employs ozone (O_3_) molecules to oxidize and decompose pollutants in wastewater leading to their elimination. Unfortunately, this method could eliminate a maximum of 80% of the pollutants and may be uneconomical due to the high equipment and operational costs [11,12,13]. Another important process is based on the phenomenon of photocatalysis, where a metal oxide catalyst is activated by absorption of photons of appropriate energy and are thus capable of speeding up the degradation reactions of phenolic compounds [14,15,16]. Despite many advantages, this method is very sensitive to changing conditions, e.g., catalyst dose, exposure time, solution pH and light intensity [17]. Another process is the Fenton reaction. It is a catalytic reaction between iron (III) and hydrogen peroxide, which leads to the formation of highly oxidative radical species including the hydroxyl radical (OH•), superoxide radical (O2^−^•), hydro-peroxyl radical (HO_2_•) and some organic radicals – alkyl (R•), peroxyl (RO_2_•) [18,19]. Those species are very effective oxidizing agents for the degradation of organic waste materials (e.g., resorcinol). In modern days the Fenton reaction is one of the most commonly used methods for the wastewater treatment. In contrast to other methods, the Fenton reaction oxidizes phenol pollutants in aqueous solution to carbon dioxide and water as by-products [20,21,22]. Despite the high efficiency and formation of harmless end products, the Fenton reaction has a disadvantage of high cost and generation of excessive volumes of sludge [23,24]. Adsorption is based on the phenomena of intermolecular forces of attraction, which occur between molecules of pollutants in wastewater and a solid phase of the adsorbent having a highly porous surface structure. As a result of this physical process, some of the solute molecules from the solution could become aggregated at the solid surface of the adsorbent [25,26,27,28]. On the other hand, the concentration of the to-be-removed substance, the presence of other organic components, variation of pH and temperature parameters can negatively affect the effectiveness of the adsorption process [29]. 

In recent years, one of the methods that has attracted a great deal of attention for the treatment of wastewater containing toxic or refractory organic pollutants is electrochemical oxidation. The utilization of this method for the degradation of phenolic compounds has many benefits, including high efficiency, easy automation and environmental friendliness [30,31,32,33,34,35,36,37,38,39,40,41]. Although numerous works have been published on this subject, only few are related to resorcinol, one of the most commonly encountered phenolic compounds.

In general, the process of resorcinol electrooxidation is associated with the conversion of a resorcinol molecule into carboxylic acids and finally the formation of H_2_O and CO_2_ molecules [33,42]. However, it is also agreed that resorcinol electrooxidation might partly lead to the formation of polymers [32]. A schematic representation and description of these processes were presented in a recent publication from this laboratory [43]. The purpose of this work is principally concerned with the kinetic aspects of resorcinol electrooxidation (degradation) and electrosorption reactions, examined on the surface of the polycrystalline platinum electrode, in 0.5 and 0.1 M H_2_SO_4_, 0.5 M Na_2_SO_4_ also 0.5 and 0.1 M NaOH supporting electrolytes by means of electrochemical methods (cyclic voltammetry and a.c. impedance spectroscopy).

## 2. Results and Discussion

### 2.1. EDX Characterization of Polycrystalline Platinum

Figure 1a,b present exemplary EDX spectra recorded prior to and after a resorcinol electrodegradation test was carried-out. Correspondingly, Table 1 presents changes in the surface composition of the polycrystalline platinum electrode, depending on the working electrolyte. The presence of 7.05 wt. % of C element on the Pt electrode surface, recorded before the measurements, most likely comes from the carbon tape used to position the sample in the sample holder. Electrooxidation of resorcinol led to a significant (solution-dependent) increase in the registered average wt. % of carbon and oxygen elements, as compared to the untested Pt electrode.

The overall significant increase of Pt surface content of carbon and oxygen elements after the resorcinol electrooxidation could be explained by the partial formation of strongly-bonded polymeric species on the surface of the platinum electrode (clearly discernible as a surface-adsorbed, yellowish layer). On the other hand, a radical increase of Pt surface oxygen content was partly caused by the platinum surface oxidation process.

### 2.2. Electrochemical Characterization of the Resorcinol Oxidation Reaction

The cyclic voltammetric behaviour of resorcinol (at 1 × 10^−3^ M) on polycrystalline Pt electrode surface is presented in Figure 2a–c for acidic, neutral and alkaline solutions, respectively, together with baseline CV profiles for solutions without resorcinol. Hence, in Figure 2a a major anodic oxidation peak can be observed in the CV profiles over the potential range 1.1–1.5 V vs. RHE (for 0.1 and 0.5 M H_2_SO_4_). This anodic peak is associated with the resorcinol oxidation process [44,45]. Nonetheless, some of the resorcinol radicals that become adsorbed on the electrode surface could partly form dimers and then polymers, causing considerable blocking of the electrochemically active area surface [10,30,31,32]. This effect is clearly visible in the CV profiles of Figure 2a, where the anodic peaks are correspondingly radically inhibited after the first cycle from 8.26 to 2.61 mA cm^−2^ and from 7.62 to 2.09 mA cm^−2^ for 0.5 M H_2_SO_4_, and 0.1 M H_2_SO_4_. Moreover, a decrease of current-density takes place with consecutive cycles. The following anodic peak starting at 1.7 V vs. RHE (for both solutions in the presence of resorcinol) is associated with oxygen evolution reaction and is significantly shifted towards more positive potentials (by *ca.* 0.2 V), as compared to baseline CV profiles. In addition, a broad cathodic peak can also be observed in Figure 2a at a potential range of 0.5–0.8 V/RHE (0.1 M H_2_SO_4_ and 0.5 M H_2_SO_4_ solutions). The latter peak most likely corresponds to desorption of OH^-^ ions [46,47]; however, it could also be associated with partial and limited reversibility of the resorcinol oxidation process on the Pt electrode surface [10,30,31,32].

Similar voltammetric behaviour was recorded on the Pt electrode, studied in 0.5 M Na_2_SO_4_, 0.1 and 0.5 M NaOH (see Figure 2b,c), where an anodic feature (associated with resorcinol oxidation) is observed over the potential ranges of 1.3–1.7 V and 1.1–1.4 V, in neutral and basic solutions, respectively. However, in contrast to the behaviour observed in acidic solutions, voltammetric profiles in neutral and alkaline solutions are characterized by radically lower current-densities, namely 3.42, 4.47 and 4.41 mA cm^−2^ for 0.5 M Na_2_SO_4_, 0.1 and 0.5 M NaOH, respectively. Nevertheless, in case of alkaline solutions, the reduction of the current-density level was less noticeable (from 4.47 and 4.41 to 3.10 and 3.02 mA cm^−2^), being actually preserved on this level (*ca*. 3.00 mA cm^−2^) after additional cycles. On the contrary, the current-density for acidic and neutral solutions, was steadily decreasing through continuous voltametric cycling. 

With respect to the peak current derived from the cyclic voltammogram for Pt electrode in 0.5 M H_2_SO_4_ solution, the results presented in this paper compare quite well with work published by Nady et al. [32], where peak current on analogues Pt electrode at a sweep-rate of 50 mV s^−1^ reached 10 mA cm^−2^ for the resorcinol concentration of 1 × 10^−2^ M.

On the other hand, the a.c. impedance behaviour of the resorcinol electrooxidation process at the polycrystalline platinum electrode, in contact with 0.1 and 0.5 M H_2_SO_4_, 0.5 M Na_2_SO_4_ also 0.1 and 0.5 M NaOH solutions, is shown in Figure 3a,b and Table 2. For all examined solutions, the Nyquist impedance spectra (along with the corresponding Bode plots) recorded over the studied potential range (1150–1250 mV/RHE for 0.5 M H_2_SO_4_ and 0.1 M NaOH, 1400–1500 mV/RHE for 0.5 M H_2_SO_4_, and 1200–1300 mV/RHE for 0.1 M H_2_SO_4_ and 0.5 M NaOH) exhibited one visible and somewhat depressed partial semicircle (or a single broad peak in the phase-angle/frequency Bode diagram). Taking into account that a small, but noticeable capacitance dispersion effect was observed on the Pt electrode surface, the CPE: constant phase element-modified equivalent circuit model with a single time constant was used to fit the experimentally obtained impedance data (see Figure 4).

The charge-transfer resistance, *R*_F_ parameter (Table 2) corresponds to the oxidation process of resorcinol on the Pt electrode surface. In case of all five examined solutions, the *R*_F_ had a tendency to increase upon augmentation of the electrode potential and reached its maximum values of 7431, 3730, 9870, 2230 and 2359 Ω cm^2^ for the solutions of 0.1 M H_2_SO_4_ (at 1300 mV/RHE), 0.5 M H_2_SO_4_ (at 1250 mV/RHE), 0.5 M Na_2_SO_4_ (at 1500 mV/RHE), 0.1 M NaOH (at 1250 mV/RHE) and 0.5 M NaOH (at 1300 mV/RHE), correspondingly. This behaviour could be explained in terms of the formation of surface-poisoning resorcinol polymer (paragraph 1 of Section 2.2.). The above phenomenon could be supported through a significant reduction of the recorded *C*_dl_ parameter values upon an increase of the electrode potential (Table 2). 

In fact, based on the recorded charge-transfer resistance values, the resorcinol electrodegradation is most efficient in alkaline solution. On the other hand, the recorded impedance data did not allow for the derivation of the adsorption charge-transfer resistance and pseudocapacitance parameters (associated with resorcinol electrosorption). However, as the *C*_dl_ parameter values recorded in Table 2 are significantly higher than those typically observed for platinum electrodes [48,49], one cannot exclude the possibility that a part of the recorded double-layer capacitance could be associated with pseudocapacitance related to surface oxidation or electrosorption phenomena.

Additionally, prolonged cyclic voltammetry trials (600 cycles, at a sweep-rate of 100 mV s^−1^ for the potential range of 500–1600 mV) were carried-out for all examined solutions in order to derive information on the mechanism of resorcinol degradation (for 1 × 10^−3^ M resorcinol). Thus, based on the results of combined HPLC/MS analysis one could assume that the process of resorcinol electrodegradation primarily leads to opening of the aromatic ring and to the formation of H_2_O and CO_2_ molecules as basic products. The above is proven by the lack of hydroquinone or benzoquinone species (or any other by-products of the resorcinol oxidation process) in the recorded chromatograms (Figure 5).

The recorded final concentration of resorcinol in 0.1 and 0.5 M H_2_SO_4_, 0.5 M Na_2_SO_4_ 0.1 and 0.5 M NaOH solutions came to 8.9 × 10^−4^, 8.7 × 10^−4^, 8.8 × 10^−4^, 7.4 × 10^−4^ and 7.8 × 10^−4^ M, respectively. Furthermore, the HPLC/MS analysis shows that electrooxidation of resorcinol proceeded faster in an alkaline environment then in acidic/neutral solutions. This is most likely caused by excessive abundance of hydroxyl species in basic medium, which could facilitate the breakdown of the aromatic resorcinol rings [5,32]. 

## 3. Materials and Methods

Supporting solutions of 0.1 M NaOH, 0.5 M Na_2_SO_4_ and 0.5 M H_2_SO_4_ were prepared from 99.99% NaOH and 99.99% Na_2_SO_4_ pellets (Merck, Warsaw, Poland), and sulphuric acid of the highest purity available (SEASTAR Chemicals, Sidney BC, Canada), respectively. All solutions were made up from ultra-pure water produced by means of Millipore Direct-Q3 UV system with 18.2 MΩ cm water resistivity. The resorcinol (>99%, Sigma-Aldrich, Warsaw, Poland) concentration was on the order of 1 × 10^−3^ M.

During the course of this study, a three-compartment electrochemical cell made from Pyrex glass was employed to carry-out the kinetic investigations of the resorcinol electrodegradation process. The cell contained three electrodes: a working electrode (WE) made from polycrystalline Pt wire (S_A_= 0.62 cm^2^: 1.0 mm diameter of 99.9998% purity, Johnson Matthey, Inc., Audubon, PA, USA) in the central part of the cell, a Pd wire (0.5 mm diameter of 99.9% purity, Aldrich) reversible hydrogen electrode (RHE) and a Pt wire (1.0 mm diameter of 99.9998% purity, Johnson Matthey, Inc.) counter electrode (CE), both in separate compartments. All procedures regarding the preparation of the electrodes and the cell pre-treatments were as those previously described in other works from this laboratory [43,50,51].

For all electrochemical measurements (a.c. impedance spectroscopy and cyclic voltammetry tests), a 12.608 W Full Electrochemical System (Solartron, Hampshire, England) was employed. All experiments were performed at room temperature. The impedance measurements were carried-out at an a.c. signal of 5 mV and the frequency was swept between 1.0 × 10^5^ and 0.5 × 10^−1^ Hz, where cyclic voltammetry measurements were performed at a sweep-rate of 50 mV s^–1^. The instruments were controlled by ZPlot 2.9 (Corrware 2.9) software for Windows (Scribner Associates, Inc., Southern Pines, NC, USA), whereas data analysis was conducted with ZView 2.9 (Corrview 2.9) software package. The impedance spectra were fitted by means of a complex, non-linear, least-squares immittance fitting program, LEVM 6, written by Macdonald [52].

Moreover, HPLC/MS analyses were performed to quantitatively evaluate the reaction products/intermediates. These analyses were carried-out by means of a HPLC (LC 20 Prominence, Shimadzu, Kyoto, Japan) system combined with a QTRAP 5500 mass spectrometer (AB SCIEX, Concord, ON, Canada), supplemented with an ESI ion source, triple quadrupole and an ion trap. Reaction products were separated by means of an XBridge C18 (3.5 µm, 150 × 2.1 mm) chromatographic column (Waters, Milford, MA, USA) at 45 °C for the mobile phase flow of 0.2 mL min^−1^. Both qualitative and quantitative analyses were conducted based on the multiple reaction monitoring (MRM) method. The quantitative analysis was performed through the application of linear calibration curves (R^2^ = 0.993), acquired by serial dilution of standard stock. A calibration curve of four points was used consisting of 3260, 1603, 326 and 32.6 μg/mL aliquots. Three replicates of all HPLC/MS measurements were performed.

Additionally, a spectroscopic characterization was performed on the Pt WE, before and after electrooxidation trials. Measurements were carried-out by means of a Quanta FEG 250 Scanning Electron Microscope (SEM, Thermo Fisher Scientific, Hillsboro, OR, USA), equipped with an Energy-Dispersive X-ray Spectroscopy (EDX) supplement (XFlash 5010, Bruker, Madison, WI, USA). The EDX analyses were derived for an acceleration voltage of 10 kV with the primary intention of confirming the presence of resorcinol-derived polymer on the surface of the examined electrode. Usually, three EDX measurements were conducted independently for all experimental arrangements.

## 4. Conclusions

The electrochemical analysis of resorcinol electrooxidation on polycrystalline platinum in 0.1 and 0.5 M H_2_SO_4_, 0.5 M Na_2_SO_4_, 0.1 and 0.5 M NaOH solutions was carried-out by means of cyclic voltammetry and electrochemical impedance spectroscopy techniques. The electrochemical oxidation of resorcinol leads to the degradation of the aromatic ring structure and partly to electropolymerization of resorcinol on the electrode surface, where generated polymeric compound(s) blocks the electrode surface, which does significantly impede electrochemical activity of the Pt electrode. The above was confirmed by EDX-derived elemental composition analysis, performed prior to and after electrochemical experiments. Although, there is no linear dependence of the solution pH on the kinetics of resorcinol electrooxidation, based on the obtained results one could easily differentiate between the electrochemical behaviour of resorcinol in all three studied solutions. Most importantly, the alkaline solution was found to appreciably facilitate the process of resorcinol electrodegradation, which was evidenced by the conducted experiments. The above most likely results from the increased surface presence of oxygen and hydroxyl species in a basic medium, as compared to that in sulphate-based, neutral and acidic solutions.

## Figures and Tables

**Figure 1 molecules-24-02309-f001:**
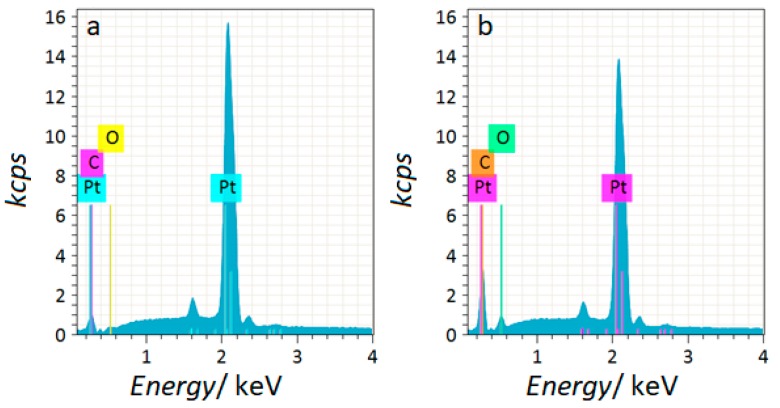
(**a**) EDX spectrum carried-out at an acceleration voltage of 10 kV and a working distance of 10.0 mm for a fresh polycrystalline platinum electrode; (**b**) as in (**a**), but after electrochemical examinations (600 cyclic voltammetry cycles-CV).

**Figure 2 molecules-24-02309-f002:**
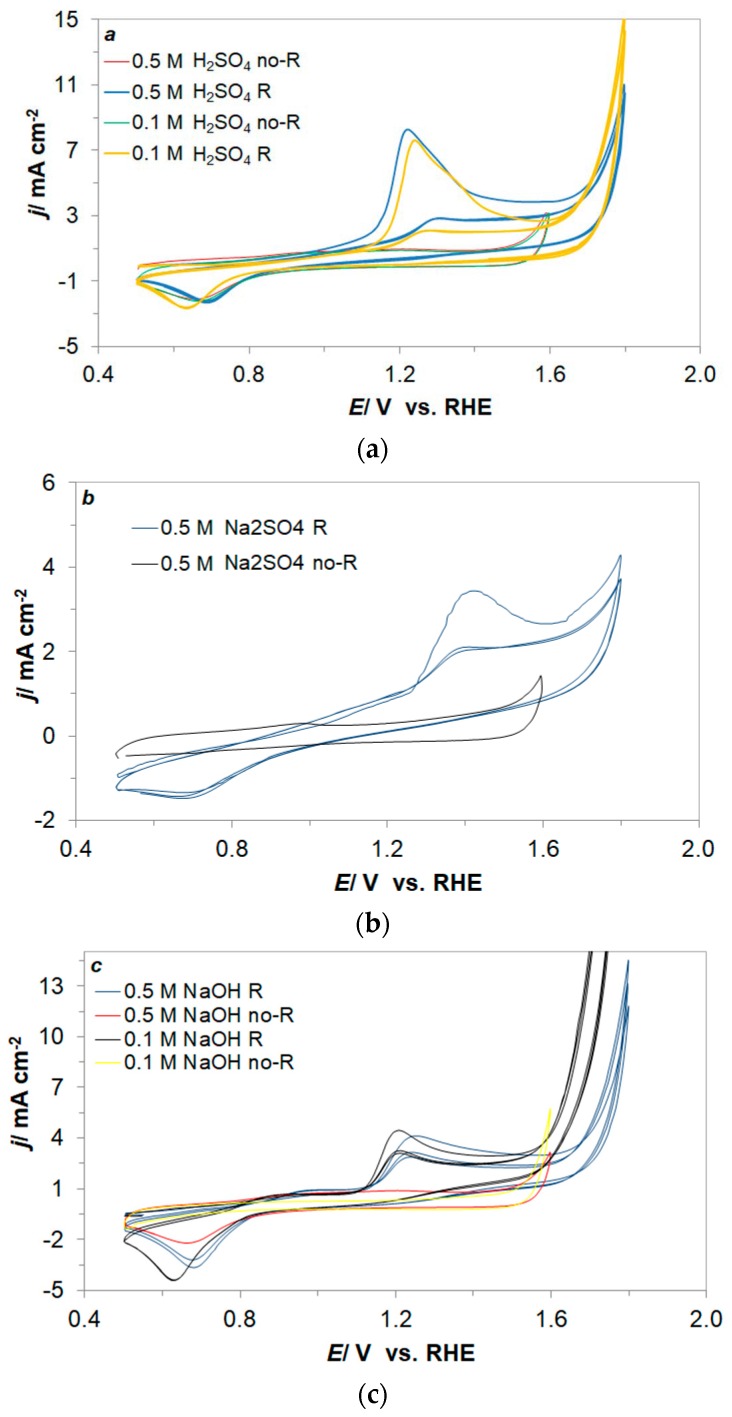
Cyclic voltammograms recorded on the polycrystalline platinum electrode surface (**a**) in 0.1 and 0.5 M H_2_SO_4_; (**b**) 0.5 M Na_2_SO_4_; (**c**) 0.1 and 0.5 M NaOH supporting solutions, carried-out in the presence (R) and absence of resorcinol (no-R) at the concentration of 1 × 10^−3^ M and a sweep-rate of 50 mV s^−1^.

**Figure 3 molecules-24-02309-f003:**
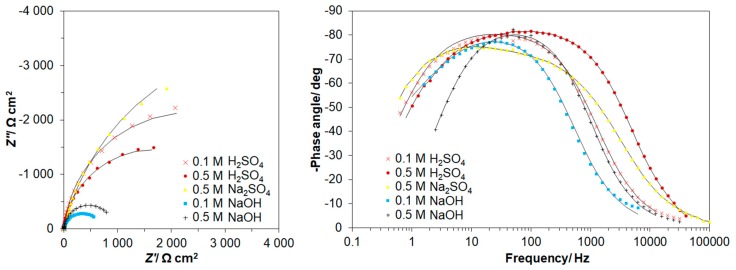
(**a**) Complex-plane impedance plots for resorcinol electrooxidation on polycrystalline platinum electrode surface, in contact with 0.1 and 0.5 M H_2_SO_4_, 0.5 M Na_2_SO_4_, 0.1 and 0.5 M NaOH solutions, recorded at room temperature for potential values from 1150 through 1400 mV vs. RHE, corresponding to the minimum values of the recorded Faradaic charge-transfer resistance parameter; (**b**) corresponding Bode impedance plots (solid lines correspond to representation of the data according to an equivalent circuit model shown in Figure 4).

**Figure 4 molecules-24-02309-f004:**
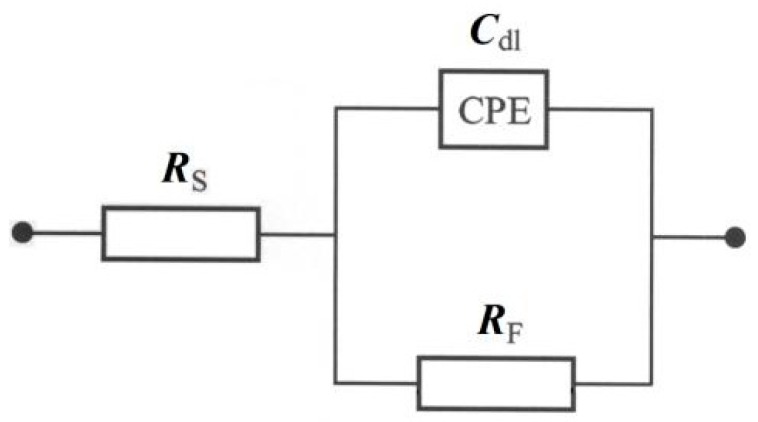
Equivalent circuit model for electrooxidation of resorcinol on Pt electrode surface. The circuit exhibits a Faradaic charge-transfer resistance, *R*_F_ in a parallel combination with the double-layer capacitance, *C*_dl_ (shown here as CPE: constant phase element), jointly in series with an uncompensated solution resistance, *R*_S_.

**Figure 5 molecules-24-02309-f005:**
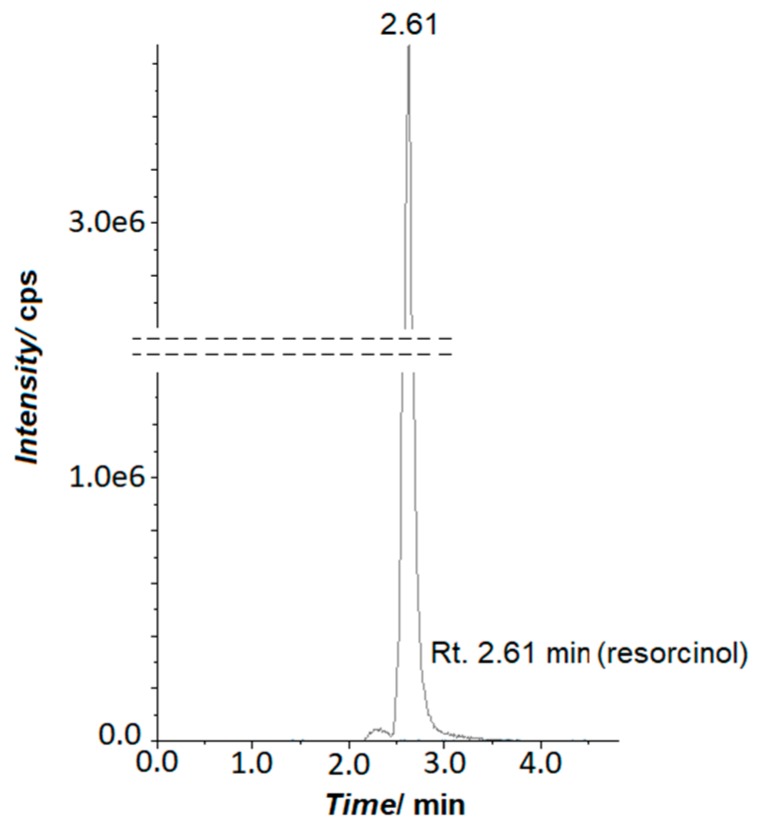
HPLC chromatogram recorded for electrochemically oxidized resorcinol in 0.5 M H_2_SO_4_ solution.

**Table 1 molecules-24-02309-t001:** EDX-derived surface elemental compositions for polycrystalline platinum samples recorded prior to and after resorcinol electrochemical oxidation (600 CV cycles), carried-out in all working solutions.

Elements	Electrolyte	Before	After
Pt/%	0.1 M H_2_SO_4_	92.50	72.54
	0.5 M H_2_SO_4_		71.93
	0.5 M Na_2_SO_4_		72.83
	0.1 M NaOH		82.73
	0.5 M NaOH		83.14
C/%	0.1 M H_2_SO_4_	7.05	24.02
	0.5 M H_2_SO_4_		24.31
	0.5 M Na_2_SO_4_		23.22
	0.1 M NaOH		15.65
	0.5 M NaOH		15.27
O/%	0.1 M H_2_SO_4_	0.45	3.44
	0.5 M H_2_SO_4_		3.76
	0.5 M Na_2_SO_4_		3.95
	0.1 M NaOH		1.62
	0.5 M NaOH		1.59

**Table 2 molecules-24-02309-t002:** Parameters for the process of resorcinol electrooxidation (at 1 × 10^−3^ M) on polycrystalline platinum electrode surface in contact with 0.1 and 0.5 M H_2_SO_4_, 0.5 M Na_2_SO_4_, 0.1 and 0.5 M NaOH solutions, achieved by fitting the equivalent circuit model shown in Figure 4 to the experimentally-obtained impedance data [dimensionless φ parameter, which determines the constant phase angle in the complex-plane plot (0 ≤ φ ≤ 1) of the CPE circuit, varied between 0.90 and 0.98].

*E*/mV	*R*_F_/Ω cm^2^	*C*_dl_/µF cm^−2^s^φ−1^
**Pt in 0.1 M H_2_SO_4_**		
1200	5018 ± 60	85 ± 6
1250	6762 ± 144	55 ± 5
1300	7431 ± 131	45 ± 3
**Pt in 0.5 M H_2_SO_4_**		
1150	2860 ± 280	91 ± 10
1200	3090 ± 360	80 ± 8
1250	3730 ± 480	79 ± 9
**Pt in 0.5 M Na_2_SO_4_**		
1400	6750 ± 430	96 ± 8
1450	8880 ± 750	93 ± 6
1500	9870 ± 970	60 ± 8
**Pt in 0.1 M NaOH**		
1150	630 ± 80	100 ± 2
1200	740 ± 90	94 ± 2
1250	2230 ± 130	64 ± 2
**Pt in 0.5 M NaOH**		
1200	945 ± 14	99 ± 5
1250	1072 ± 34	82 ± 3
1300	2359 ± 49	75 ± 3

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
