# Peer review of "The Influence of Solution pH on the Kinetics of Resorcinol Electrooxidation (Degradation) on Polycrystalline Platinum"

_molecules, 2019, doi:10.3390/molecules24122309_

Round 1

Reviewer 1 Report

The  manuscript now is ready for publication, please, check the english language again.

Author Response

The English language has now been checked.

Reviewer 2 Report

The authors have compared the electrochemical oxidation of resorcinol on Pt in various media and made conclusion on full oxidation of the resorcinol to CO2 and water with reference to HPLC result. Although the experiments are rather well described, I am strictly against such a conclusion. Voltammograms are rather simple and do not contain any evidences of such a full oxidation. The use of conventional electrochemistry tools would clarify some aspects of the reaction including ratio of electrons and protons transferred and possible stages of reaction. Then, a scale electrolysis would give more information on the products of electrooxidation of resorcinol. Some other comments:

Page 1, line 36: “They are classified as the most toxic industrial pollutants” – do not take to the extreme, phenols were never considered as most dangerous pollutants (see Stockholm convention and dirty dozen of persistent organic pollutants).

Page 1, line 23: Fenton reaction generates not only hydroxyl radicals but also superoxide anion radicals, please add some additional references about this reaction

Page 5, line 150: the number of cycle should be marked on the Figure 1 for each electrolyte

Table 2: R(F) of 67626 Ohm /cm2 is not explained in the text – misprint?

Author Response

In earlier works from this laboratory, we were able to identify small amounts of by-products from phenols degradation process [1-3]. Thus,      while we were unable to detect any hydroquinone or benzoquinone species or any other by-products of the resorcinol oxidation process we had suggested full degradation to CO2 and water. While we understand and partly agree with the Reviewer that it would require additional investigation to determine possible stages of this reaction, we decided not to include more experiments as it was not the aim of this work.

We have now changed the implication that phenols are the most toxic industrial pollutants,      see the revised manuscript.

In the revised manuscript, we have now added some additional information and references for the Fenton reaction.

According to the Reviewer’s suggestion, we have included the number of cycles in Figure 1.

The RF value of 67626 Ω cm2  happened to be incorrect, in the revised manuscript it has been changed to the value of 6762 Ω cm2.

Round 2

Reviewer 2 Report

I agree with the changes made in accordance with Reviewer requirements and believe the manuscript can be published in present form

This manuscript is a resubmission of an earlier submission. The following is a list of the peer review reports and author responses from that submission.

Round 1

Reviewer 1 Report

This work presents some interesting work on the effect of pH in electrooxidation processes. In my opinion, the paper needs major revision before acceptance. 

(1) Electrochemical measurements only provide information about organics oxidation in the potential region before water diacharge, the degradation experiments (with resorcinol and TOC removal percentages as responses) should be performed and the results should be discussed. Specifically, the effects of influencing factors (at least four) should be checked and discussed.A modern experimental design is recommended.

(2) More supporting electrolytes should be employed, such as 0.1 M H2SO4 and 0.5 M NaOH.

(3) Most of the references are rather old, the new and recent work should be cited and discussed.

Reviewer 2 Report

The manuscript titled “The Influence of Solution pH on the Kinetics of Resorcinol Electrooxidation on Polycrystalline Platinum” can be accepted for publication after the following minor revisions.

1.       Add more details (couple of paragraphs) in the introduction section about the influence of resorcinol on the environment, different techniques used for their removal. Drawbacks of the conventional purification methods. Importance of advanced electrochemical oxidation etc.

2.       Figure 2: Use different colors for CVs under different pH conditions.

3.       Figure 3: Use different colors for EIS under different pH conditions

4.       Do an overall English editing of the manuscript

5.       Include images of the Pt electrodes in Figure 1. Is there any change in surface morphology with change in pH?.

Reviewer 3 Report

In the introduction the aim of the work is missing. Consequently, the all manuscript contains not connected experimental parts and results.

It is in accordance that the title of the manuscript and conclusion are not in a proper connection.

The experimental part is given without necessary details  (HPLC/MS analysis). It can be easily improved.

The serious mistakes are made during the cyclic voltammetry experiments and the results presentation.

The presented results concerning cyclic voltammetry make no sense. Cyclic voltammograms are confusing and not clear and  additionally they are not correct. Beside, the potential scale is not correct.  The  cyclic voltammograms of platinum electrode without resorcinol must be provided  for the all  three electrolytes  and number of cycles must be assigned . The  confusing Fig 2 must be divided in 3 separate figures (for acid, alkaline and neutral electrolyte).

 In a conclusion it is noticed that alkaline solution is best one for the degradation process. It is very interesting because the cyclic voltammetry  (Fig2) shows no any reaction of resorcinol on platinum electrode in alkaline solution. In the title of the manuscript the degradation is not mentioned at all.